# Dynamic Analysis and Path Planning of a Turtle-Inspired Amphibious Spherical Robot

**DOI:** 10.3390/mi13122130

**Published:** 2022-12-01

**Authors:** Liang Zheng, You Tang, Shuxiang Guo, Yuke Ma, Lijin Deng

**Affiliations:** 1School of Electronic Information Science and Technology, Jilin Agricultural Science and Technology University, Jilin 132101, China; 2Key Laboratory of Convergence Medical Engineering and System and Healthcare Technology, the Ministry of Industry Information Technology, School of Life Science, Beijing Institute of Technology, Haidian District, Beijing 100081, China; 3School of Artificial Intelligence, Changchun University of Science and Technology, Changchun 130022, China

**Keywords:** amphibious spherical robots, path planning, thruster evaluation, underwater motion, diamond parallel leg

## Abstract

A dynamic path-planning algorithm based on a general constrained optimization problem (GCOP) model and a sequential quadratic programming (SQP) method with sensor input is proposed in this paper. In an unknown underwater space, the turtle-inspired amphibious spherical robot (ASR) can realise the path-planning control movement and achieve collision avoidance. Due to the special underwater environments, thrusters and diamond parallel legs (DPLs) are installed in the lower hemisphere to realise accurate motion control. A propulsion model for a novel water-jet thruster based on experimental analysis and a modified Denavit–Hartenberg (MDH) algorithm are developed for multiple degrees of freedom (MDOF) to realize high-precision and high-speed motion control. Simulations and experiments verify that the effectiveness of the GCOP and SQP algorithms can realize reasonable path planning and make it possible to improve the flexibility of underwater movement with a small estimation error.

## 1. Introduction

A turtle-inspired amphibious spherical robot (ASR) is an attractive research topic for a wide range of applications in complex marine environments [1,2,3,4,5,6,7]. These robots can be applied to pollution detection and to scouting potential approach lanes for amphibious naval operations in constricted areas [8]. The amphibious spherical robot proposed by the present design is an advanced-execution device that includes a motion thruster, sensors, control boards, a communication model, a balance device, and a lithium battery installed in the upper hemisphere [9,10,11,12,13,14,15]. As a small robot, spherical robots have a wide range of applications that mainly rely on four technical advantages. The first advantage is that the spherical robot is a mobile robot that walks in a rolling manner and can maintain an advanced balance and stability, dynamic conditions with stable motion, a fast-moving speed, and a strong steering ability. The second advantage is that the spherical robot has a better seal and can completely protect the internal control unit and mechanism, which is not possible for other robots. The third advantage is that the spherical robot has vintage adaptability and can run in an unmanned area, such as a dusty, moist, corrosive, and harsh environment. The last advantage is that the spherical robot has low energy consumption, which can make the robot work longer.

Propulsion devices play a vital mechanical role in amphibious robots, achieving wide applications in complex environments [16]. With the development of spherical robot technology and its applications, many kinds of propulsive devices have been proposed and developed, such as wheel-propeller fins [17], fish-propeller robots [18], repetitive leaping robotic dolphins [19], and curved flipper legs [20]. In [21], a comparison between the computational fluid dynamics (CFD) value of the thruster and the thrust obtained in the previous thrust experiment was obtained, which improved the thrust efficiency. The prediction thruster of a propeller was studied in [22], where experimental results were provided under static and dynamic conditions. The thrust prediction of a propeller-rudder system was conducted in the cylinder section of a cavitation tunnel [23]. A bollard pull test of a ducted thruster was carried out to predict the thrust when the blades rotated in a forward direction and in the inverse direction [24]. In [25], a hydrodynamic analysis-based modelling and experimental verification of a new water-jet thruster was proposed to verify that the new thruster would improve the stability and flexibility of the amphibious robot. Therefore, thrust is simultaneously affected by the motor model, propeller design, and hydrodynamic factors [26]. In this paper, we propose a novel water-jet thruster structure installed in the lower hemisphere of an amphibious spherical robot (ASR) that has higher flexibility and maintains better underwater movement and noise interference. Through a predesigned measuring experimental platform, real-time electronic sensors were used to collect the thrust power provided by the propeller and provide the final measurement data. The measurement result shows that the dynamics, stability, and underwater movability of the ASR were improved. Thereafter, the simulation and experimental results were utilised to identify the unknown parameters of the theoretical model.

The robot performs complex tasks, encountering obstacles and using effective algorithms to demonstrate that robots designed to avoid obstacles and select a reasonable path can improve the efficiency of ASR underwater movement, as well as avoid damage or disappearance. In [27], a fuzzy control method that incorporates multisensor technology to guide underwater is presented. A dynamic path-planning method based on a gated re-current unit-recurrent neural network model is introduced for the path planning problem of a mobile robot in an unknown space [28]. Bae proposed a noble multirobot path-planning algorithm using deep learning combined with a convolution neural network (CNN) algorithm [29]. A path planning and control approach of a nonholonomic three-wheeled mobile robot (WMR) for online navigation in a road following and roundabout environments is presented in [30]. In [31], a method called the reinforced rim jump (RRJ) was developed that does not rely on point-by-point traversal; it obtains the shortest path by finding the tangent multiple times between obstacles. The abovementioned research demonstrated the excellent performance and great potential of path planning in promoting technological innovation for underwater robots. Although many types of research have been performed on path planning for underwater robots, most of the research is based on the path planning of terrestrial robots and algorithm verification using simulation experiments. In this paper, we propose the general constrained optimisation problem (GCOP) and sequential quadratic programming (SQP) algorithms for underwater circumstances and use path-planning algorithms to avoid underwater spherical obstacles with the path-planning scheme. This study sets the foundation for practical performance and enables robots to perform more complex underwater missions.

On the basis of the aforementioned analyses, we mainly focus on developing an appropriate path-planning algorithm and water jet thrust to allow a bioinspired robot to achieve more effective underwater movement. As a result, a comprehensive analysis combining the dynamic aspects is performed to facilitate the model simplification and the implementation of the control algorithm. Meanwhile, a novel improved SQP algorithm based on GCOP is presented by the proposal of the path-planning strategy. To further increase flexibility, based on preliminary research [32,33,34], we propose a new type of diamond parallel leg (DPL) to improve stability and efficiency, regardless of whether the robot is walking on land or moving underwater. In brief, the major contributions serve as an alternative to the development of a bioinspired amphibious robotic platform, allowing for the development and reliability of each method. More importantly, a repetitive DPL structure with high speed and manoeuvrability involves a better mechanical structure-optimisation and stable motion control refinement; related competitive simulations and experiments have implications for the improvement of future aquatic amphibious robots.

The rest of this paper is organised as follows. In Section 2, we describe the general mechanical design of the amphibious spherical robot. In Section 3, the formation-control modelling and the associated algorithm comparison are presented. In Section 4, we introduce the improved water-jet thruster and the leg mechanical structure and use experiments to prove the stability of the structure. We perform the most important related experiments with the communication and collaboration modules regarding the optimal control strategies and evaluate the new structure in Section 5. Finally, the conclusions and future work are presented in Section 6.

## 2. Mechanical Design

For this paper, we designed a novel turtle-inspired amphibious spherical robot (ASR) which could move on land and underwater to perform complicated tasks. The robot had two actuating modes: quadruped walking mode and water-jet propulsion mode [35,36], as shown in Figure 1, and the structure of the ASR was moderate. The ASR also had the ability to move from the ground to underwater. The best designation of the amphibious robot could alter the drive mode between the water-jet propulsion mode and quadruped walking mode with the structure of complicated propulsion mechanisms.

The structure of the amphibious robot was composed of two parts that contain a sealed transparent upper hemisphere and transparent quarter-spherical shells. The shell could be opened and closed through the main circuit board and sent a digital signal to control the two servo motors. Between the upper hemisphere and the lower hemisphere, a plastic plate was used to carry the circuit boards, four actuating leg units, and the battery installed on this plate, as shown in Figure 2. The leg structure is shown in Figure 3, and the moving distance of each step distance was 85 mm, as shown in Figure 4.

To estimate the flexibility and speed of the amphibious spherical robot (ASR), we needed to redesign a hydrodynamically stable underwater thruster. The new water-jet thruster with a conical nozzle was an electromechanical device equipped with a motor and a propeller wrapped by a cylinder duct. The material of the propeller was carbon fibre made by a three-dimensional (3-D) print, which meant that the dynamics of the new thruster could be simplified. The streamlined shape of the thruster also reduced the resistance; the resistance received in the water would be made small, and the thrust generated would be greater. The new thruster mainly consisted of five parts: a waterproof motor, a propeller, a motor bracket, an anti-skid device, and a conical nozzle, as shown in Figure 4. The propeller had five blades, and the conical nozzle produced higher thrust than a cylindrical nozzle. The genius of the design was that at the bottom of the water jet, an anti-skid device was installed on the bottom of the leg to increase friction and ensure that the robot could walk on land more stably.

## 3. Control Modelling

This section mainly introduces diamond parallel leg (DPL) control methods, using Denavit-Hartenberg (DH) and modified-Denavit-Hartenberg (MDH) algorithms to model and analyse the DPL. The MDH algorithm is an improvement of the control function on the DH algorithm. The traditional DH is the basic dynamic modelling method; it is generally either two-link or multilink modelling. The dynamic-control method of the MDH algorithm proposed in this paper is based on the synthesis of the motion state of a couple of two-links to control a transmission link. There are fundamental differences between modelling and equations of motion.

This section also introduces modelling and analyses the general constrained optimisation problem (GCOP) and a sequential quadratic programming (SQP) algorithm. The SQP algorithm is mainly based on the analyses of the original- and dynamic-displacement coordinates, combined with the transition matrix and position information to implement and control the movement path. The GCOP algorithm calibrates and restricts the coordinates of obstacles under a fixed-path trajectory. The advantage of the SQP algorithm is that it can plan and select a path. The GCOP algorithm calibrates obstacles under a fixed path. There are essential differences between the two motion-control algorithms.

Figure 5 is a modelling calibration of the MDH algorithm. The joint of the diamond parallel leg (DPL) rotates on a single axis, and most angle cosine values are special values (zero or one). The posture description and coordinate-transformation equation can be expressed as follows:(1)$$T_{n}^{0}\left(q\right)=A_{1}^{0}\left(q_{1}\right)A_{2}^{0}\left(q_{2}\right)\ldots A_{n}^{n-1}\left(q_{n}\right)$$
where *q* is the vector of the joints, which requires a recursive calculation of positive kinematics equations, and the synthesis of the transformation matrix is $A_{i}^{i-1}$(*q_i_*), (*i* = 1, 2,…, *n*). Each homogeneous-transformation matrix is a function of a single joint variable according to positive calculation. Describing the actual transformation of the position and the direction of the end-effector coordinate system relative to the base coordinates is as follows:(2)$$T_{e}^{b}(q)=T_{0}^{b}(q)T_{n}^{0}(q)T_{e}^{n}$$
where $T_{e}^{b}$ and $T_{e}^{n}$ are homogeneous transformations used to describe the position and direction of coordinate system relative to the base coordinate system, respectively. They also describe the movement status information of the effector relative to the end coordinate.

Due to the symmetrical structure of the parallel legs, the homogeneous transformations of 1′, 2′ and 1″, 2″ to 3 are equivalent, and we can conclude that
(3)$$A_{i}^{i-1}\left(v_{i}^{\prime }\right)=A_{i}^{i-1}\left(v_{i}^{\prime \prime }\right)=\left[\begin{array}{cccc} c_{i} & -s_{i} & 0 & a_{i}c_{i}\\ s_{i} & c_{i} & 0 & a_{i}s_{i}\\ 0 & 0 & 1 & 0\\ 0 & 0 & 0 & 1 \end{array}\right]$$
where *c_i_* = cos *θ_i_*, *s_i_* = cos *θ_i_*, and *a_i_* is the node distance, and the coordinate transformations of the two branches of the parallelogram leg are as follows:(4)$$A_{2^{\prime }}^{0}\left(q_{1}^{\prime }\right)=A_{1^{\prime }}^{0}A_{2^{\prime }}^{1^{\prime }}=\left[\begin{array}{cccc} c_{12} & -s_{12} & 0 & a_{2}^{\prime }c_{12}+a_{{}_{1}}^{\prime }c_{1}\\ s_{12} & c_{12} & 0 & a_{{}_{2}}^{\prime }s_{12}+a_{{}_{1}}^{\prime }s_{1}\\ 0 & 0 & 1 & 0\\ 0 & 0 & 0 & 0 \end{array}\right]$$
(5)$$A_{2^{\prime }}^{0}\left(q_{2}^{\prime }\right)=A_{1^{\prime \prime }}^{0}A_{2^{\prime \prime }}^{1^{\prime }}=\left[\begin{array}{cccc} c_{12} & -s_{12} & 0 & a_{{}_{2}}^{\prime \prime }c_{12}+a_{{}_{1}}^{\prime \prime }c_{1}\\ s_{12} & c_{12} & 0 & a_{{}_{2}}^{\prime \prime }s_{12}+a_{{}_{1}}^{\prime \prime }s_{1}\\ 0 & 0 & 1 & 0\\ 0 & 0 & 0 & 0 \end{array}\right]$$

According to Equations (1) and (2), the positive kinematics equation can be derived as follows:(6)$$T_{3}^{0}\left(q\right)=A_{2^{\prime }}^{0}\left(q_{1}^{\prime }\right)A_{3}^{3^{\prime }}=\left[\begin{array}{cccc} c_{12} & -s_{12} & 0 & a_{4}c_{12}+a_{2}^{\prime }c_{12}+a_{1}^{\prime }c_{1}\\ s_{12} & c_{12} & 0 & a_{4}s_{12}+a_{2}^{\prime }s_{12}+a_{1}^{\prime }s_{1}\\ 0 & 0 & 1 & 0\\ 0 & 0 & 0 & 1 \end{array}\right]$$

### 3.1. SQP (Sequential Quadratic Programming) Algorithm

To model and operate an amphibious spherical robot (ASR) in three-dimensional movement, the spatial position, direction, size, and shape of water surroundings must be constructed by six independent variables [37].
(7)$$x=x(x_{0},y_{0},z_{0},\theta_{1},\theta_{2},\theta_{3},v)$$
(8)$$y=y(x_{0},y_{0},z_{0},\theta_{1},\theta_{2},\theta_{3},v)$$
(9)$$z=z(x_{0},y_{0},z_{0},\theta_{1},\theta_{2},\theta_{3},v)$$

Among them, Equations (8)–(10) are vectors that construct the robot’s motion-control trajectory, three-axis motion displacement vector, three-axis rotation vector, and the robot’s boundary-point vector. *x*_0_, *y*_0_, *z*_0_ are the spatial positions, *θ*_1_, *θ*_2_, *θ*_3_ are the Euler angles, and *v* is a vector with two parameters (*t*_1_, *t*_2_) that are used to represent the boundary point of a specific robot. The spherical robot is defined as follows:(10)$${\left[{\left((x-x_{0})/r_{x}\right)}^{2/s_{2}}+{\left((y-y_{0})/r_{y}\right)}^{2/s_{2}}\right]}^{s_{2}/s_{1}}+{\left((z-z_{0})/r_{z}\right)}^{s/s_{1}}=1$$
where $x=r_{x}\cos^{s_{1}}(t_{1})\cos^{s_{2}}(t_{2})$, $y=r_{y}\cos^{s_{1}}(t_{1})\cos^{s_{2}}(t_{2}),\;-\pi /2\leq t_{1}\leq \pi /2$, $z=r_{z}\cos^{s_{1}}(t_{1}),\;0\leq t_{1}\leq 2\pi$.

Assuming that the Cartesian coordinates of an object are (*x_old_*, *y_old_*, *z_old_*), using the rotation matrix *R*, the equation of motion from the old coordinates to the new coordinates can be derived. The (*x_new_*, *y_new_*, *z_new_*) coordinate is expressed as follows:(11)$$\left(\begin{array}{l} x_{new}\\ y_{new}\\ z_{new} \end{array}\right)=R\left(\begin{array}{l} x_{old}\\ y_{old}\\ z_{old} \end{array}\right)$$

The path-planning problem is transformed into a semi-infinite constrained-optimisation problem. Based on the SQP algorithm, assuming that there are many obstacles in the space, the surfaces of the definite obstacle can be expressed as follows:(12)$$h_{j}(x,y,z)=1,j=1,2,\cdots ,n$$

The entire free space can be represented by the following inequality, and finally simplifies to
(13)$$1-h_{j}(x,y,z)\leq 0,j=1,2,\cdots ,n$$

The necessary and sufficient condition for avoiding collision is that all points on the surface must be collision-free. The sufficient and essential condition for becoming collision-free is as follows:(14)$$1-h_{j}\left(x_{loc},y_{loc},z_{loc}\right)\leq 1$$
where
(15)$$\begin{array}{l} x_{loc}=x\left(x_{0},y_{0},z_{0},\theta_{1},\theta_{2},\theta_{3},v\right)\\ y_{loc}=x\left(x_{0},y_{0},z_{0},\theta_{1},\theta_{2},\theta_{3},v\right)\\ z_{loc}=x\left(x_{0},y_{0},z_{0},\theta_{1},\theta_{2},\theta_{3},v\right) \end{array}$$

### 3.2. GCOP Algorithm

The surface of an object is represented by the equation *h_i_*(*x*), *i* = 1, 2,…, m, and the restricted equation is as follows:(16)$$v_{i}={\left(h_{i}^{2}+t^{2}\right)}^{1/2}+h_{i}$$
where $v_{i}$ is a small positive real number and $v_{i}$ is a function of t.

For underwater-robot path-planning, the optimal path should maintain an effective distance from obstacles. Therefore, a small positive number Δv is introduced as the distance-control parameter, and the inequality is satisfied as follow
(17)$$V=\sum v_{i}\geq \Delta v,\;\Delta v-\sum v_{i}<0$$

This point must be outside of the obstacle determined by (18). If Δv→0, the boundary determined by $\Delta v-\sum v_{i}\leq 0$ will approach the surface of the obstacle. If the surface and the exterior are determined by $\left(h_{1}\geq 0)\vee (h_{2}\geq 0)\vee \cdots \vee (h_{m}\geq 0\right)$, then its exterior and surface can also be determined by $\Delta v-\sum v_{i}\leq 0$, where Δv→0 and $\Delta v-\sum v_{i}\leq 0$ are satisfied, and then it falls outside of the object.

## 4. Experimental Verification

### 4.1. Experimental Setup

To evaluate the performance of the novel thruster module (Figure 6) and the new structure of the diamond parallel leg (DPL), some necessary experiments are conducted in this section. A tank was prepared with water at a depth of 700 mm, and a measuring platform based on an electronic control system and pressure sensor was employed to measure the propulsive force of the multivectored water-jet thruster. One power supply was installed as the electric source and provided a 7.4 V (volt) power to a vectored water-jet thruster. The experimental setup is shown in Figure 7. All measuring modules and devices connected with the pressure sensor by using the aluminium alloy profile could be considered rigid connections.

The variety of the underwater environment is less than that of the air, and the propulsive force is improved and relatively easily to implement. The speed of the amphibious spherical robot (ASR) can be enhanced. When the robot moves at a constant speed, the acceleration is equal to zero. It can be inferred that the movement process is the initial stage of acceleration, and the propulsion force will be greater than the resistance. The actual relationship between thrust and resistance is to determine the energy required to make the ASR achieve more stable movement. The data-acquisition module was designed by using a 24-bit analogue-to-digital converter (ADC) chip and was then displayed by a 1602 liquid-crystal display (LCD). The water jet needed to be in full contact with water and not touching the air. After adjusting the direct-current (DC)-regulated power supply to 4.5 V, we found that the water jet could not be activated, and by turning the voltage supply to 5 V, the motor began to rotate slowly. When the voltage gradually increased to 7.4 V, the current gradually approached 3 A (amperes), and the sensor display was 17 N (Newton). We continued to raise the voltage; when the voltage reached 7.9 V, the current was saturated, and the red light indicated that the motor was blocked when the switch for the power supply started to flash. Starting from 7.4 V, the value measured by the pressure sensor was approximately 17 N, and the data changes are shown in Figure 8. The force was taken by the new water jet at a fixed moving time compared with the previous water jet, which proves that the new water jet could generate more thrust and enabled the robot speed to also greatly improve. The current thruster velocity was increased by 300 mm/s from the previous thruster of 180 mm/s within 0−20 s [38].

In the modified-Denavit-Hartenberg (MDH) algorithm’s motion-control experiments, we also used a diamond parallel platform that included a diamond parallel leg (DPL), two servo motors, an experiment platform, an electronic control board, a servo motor driver board, a power supply, and an LCD to calculate the tension of the new leg structure and find the best distance between the centre points of the two servo motors, as shown in Figure 9. While MDH was implemented on the testbed, the DC servo motors were set up to work under “torque output mode”. First, the servo motor drive plate was used to turn the steering gear on the right side of the leg structure from the extreme position of 45° to 90°. To make the experimental process clearer, the steering speed of the steering gear was adjusted from 180°/s to 45°/s. Through the display screen, we could clearly observe that as the angle of the rocker leg of the steering gear decreased, the number of indications gradually increased from the original value of 1.2 Nm to 102 Nm. As shown in Figure 10, it was proven by the experiments that a steering gear of a single leg rotating 45° could generate 102 Nm thrusts. There were two servos on each diamond parallel leg, and each leg could generate approximately 204 Nm thrusts. The ASR had four drive legs, so it could generate 816 Nm thrusts. It is known that all of the legs could support an object with a gravity of 816 Nm; that is, the four legs could support a weight of 81.6 kg (kilograms). In practical applications, the moving process of the diamond parallel usually performed a lift *X* stretching direction. As shown in Figure 11, each moving stage was further divided into two equal distance moving speeds: a constant acceleration stage and a constant deceleration stage.

To estimate the speed and position trajectories, some specification parameters were defined. With the MDH control algorithm, the torque could reach peaks of 102 Nm and -102 Nm. Without the MDH algorithm, the collected torque data were unstable, which could lead to insufficient support of movement, as shown in Figure 12.

### 4.2. Path-Planning Evaluation

The effectiveness of the constrained-optimisation problem (GCOP) model and the sequential quadratic programming (SQP) algorithm lay in the path-planning procession of an amphibious spherical robot (ASR). MATLAB software was used to set the scene-range of the simulation experiment from the initial starting coordinates of (−200, −200, −200) to the target of (500, 500, 500). The scene space was in a square shape with a length, width, and height of 600 centimetres. 

Four spherical obstacles were set in the simulation experiment to determine the accuracy of the SQP-based algorithm. To obtain optimised experimental results, the coordinates and equations of the four obstacles were defined; the simulation results are shown in Figure 13, and there were four obstacles between the start and terminal points. The pink curve represents the path of the SQP algorithm, and the grey curve represents the path of the GCOP algorithm. In the experiment, three sampling points were set to evaluate the two path-planning algorithms. Based on the GCOP algorithm, the obstacles were closer to the obstacle at (−100, 190, 500) and (−5, 200, 345.8), while the SQP algorithm corresponded to the coordinates that were relatively far from the obstacle. At the end-coordinate position, the estimated endpoint coordinate based on the GCOP algorithm deviated from the preset terminal point. The SQP algorithm basically satisfied the requirement of optimal path-planning, and the coordinate deviation of the two sampling points was optimised so that the distance from the obstacle was within a reasonable range. The SQP algorithm had a more reasonable obstacle distance at the two collection points. At the third sampling point, the deviation trajectory of the two algorithms was 167.5 mm.

## 5. Underwater Experiments

In this section, we implemented the proposed path-planning algorithms into the image-based trajectory path-planning task. The experimental analysis under the environment of a square tank that was 4000 mm in length, 2000 mm in width, and 800 mm in height with a water depth of 700 mm is shown in Figure 14. The experiment employed sequential quadratic programming (SQP) to verify the rationality and effectiveness of the control algorithm by using the image-acquisition OpenMV (open machine vision) module and the blue-light sensor to avoid obstacles, which allowed the ASR to move from the start position to the terminal position in the optimal path.

Based on the aforementioned consideration, the experimental procedures were divided into two steps. The first step was the robot bypassing the obstacle to reach the destination, as shown in Figure 15. The second step was the robot passing through the obstacle. The snapshot sequences were extracted from a video recorded live without any modification, which depicted a laser-ranging module, and the image-acquisition sensor collected real-time coordinates of obstacles in the water to adjust the motion trajectory.

Figure 15a shows the initial position at *t* = 0 s. The red arrow indicates the planned optimal path. Figure 15b–i are the actual trajectories moving from t = 0 s to *t* = 15 s. During this experiment, six different time points were collected to the status data of the moving process, and the displacement changes in the *y*-axis and *z*-axis directions were sampled to determine the stability and reliability on the selection of the optimal path. Figure 16 and Figure 17 show that the robot had a large offset error at the second and third sampling points with upper and lower error values of 20 mm and less than 10 mm, respectively, at the remaining four sampling points. The main source of error from the second to third sampling points was mainly the stage of path selection. This effect may come from the fact that the robot was rotated due to the impact of the water at a certain time so that the sensor was in a position without finding the obstacle, which inevitably prevented the sensors from receiving the distance data immediately, and the ASR had no ability to adjust the moving direction. When setting obstacles, coloured balloons filled with water which could be used to ensure the depth and position of the obstacles. Colour information could also be collected by the OpenMV sensor. A transparent spherical shell was set on the outside so that the laser sensor could detect the distance information. The diameter of the yellow obstacle was 400 mm, and the remaining obstacle diameter was 200 mm. In addition, the experiment used the SQP algorithm and GCOP algorithm to conduct three comparative experiments of path-planning algorithms. As shown in Figure 18, the SQP algorithm was more reasonable than the GCOP algorithm on the optimal path planning, and in terms of obstacle judgement, the SQP algorithm kept 200 mm from the obstacle, but the GCOP was close to the obstacle, and there was a possibility of colliding with obstacles.

Compared with the first moving-state experiment, as shown in Figure 19, the second part of the obstacle-crossing experiment was comparatively unreasonable, and the errors in the two directions of the *y*-axis and *z*-axis were relatively large. The maximum error reached approximately 150 mm, which occurred at the second sampling point in the *z*-axis movement direction, as shown in Figure 20 and Figure 21. Under different obstacle settings, as shown in Figure 22, the GCOP algorithm still had the possibility of colliding with obstacles. The advantage of the GCOP algorithm was the calibrations of the obstacles’ coordinates. In terms of path planning, the parameters of the robot’s path-planning could not be updated in time, and the transition matrix of the new motion coordinates could not be changed with the update of the motion trajectory. It was verified that the SQP algorithm is more effective than the GCOP algorithm on the efficiency of path planning.

## 6. Conclusions

A novel amphibious spherical robot (ASR) with a more stable propulsion module and better performance than a previous version was developed. First, a new propulsion thruster is proposed to improve the dynamic performance and flexibility of the ASR, and the maximum thrust is increased from the previous 8.7 N to 17 N, which allows the robot to carry more sensors without compromising stability. Second, by designing an experimental platform with a pressure sensor, the linear relationship between the thrust and voltage of the water jet is measured in real time, thus laying a theoretical foundation for realising underwater movement and path-planning.

To ensure fast, accurate, and stable cooperation to control the movement of the ASR, in this paper, we discuss developing a leg structure based on a diamond parallel construction to provide greater movement speed and stable support. A modified Denavit-Hartenberg (MDH) control algorithm is designed based on the dynamic model. Through the experiment of the pulling force, combined with the control algorithm, it is concluded that this diamond parallel leg (DPL) structure can support a maximum load of 81.6 kg and a maximum torque of 102 N. Therefore, the proposed MDH is an effective and practical control method for the cooperating motion of the diamond parallel leg. More importantly, this paper also proposes a path-planning algorithm based on the SQP algorithm and compares it with the GCOP algorithm. Combined with a modelling analysis, ASR can realise obstacle avoidance and achieve optimal path-planning for underwater obstacles through simulations and underwater experiments. The path-planning for four spherical obstacles lays the technical foundation for the ASR to achieve more complex and difficult underwater missions.

For future research, continuous improvement efforts on the mechanical design and control approaches will be devoted to the optimisation of multirobot cooperation. In addition, the theory of models that can manage the dynamics system with multiple degrees of freedom to control multiple robots to achieve rounding and hunting is worthy of investigation.

## Figures and Tables

**Figure 1 micromachines-13-02130-f001:**
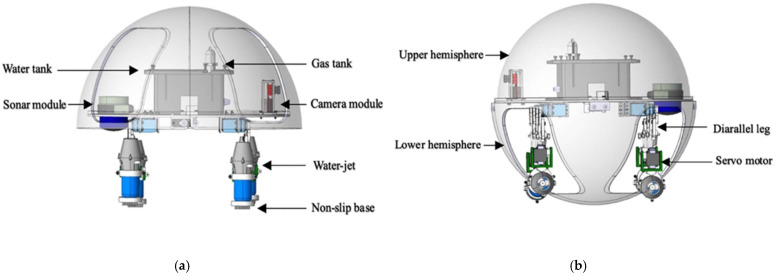
Structure of the amphibious spherical robot. (**a**) Land posture; (**b**) underwater posture.

**Figure 2 micromachines-13-02130-f002:**
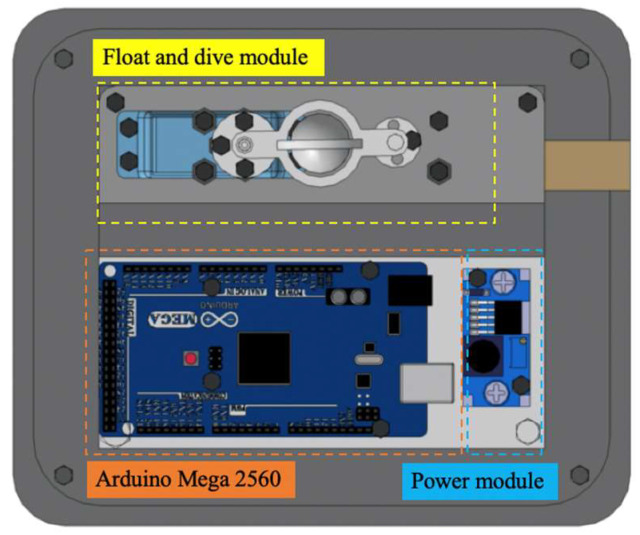
Interior layout of the waterproof tank.

**Figure 3 micromachines-13-02130-f003:**
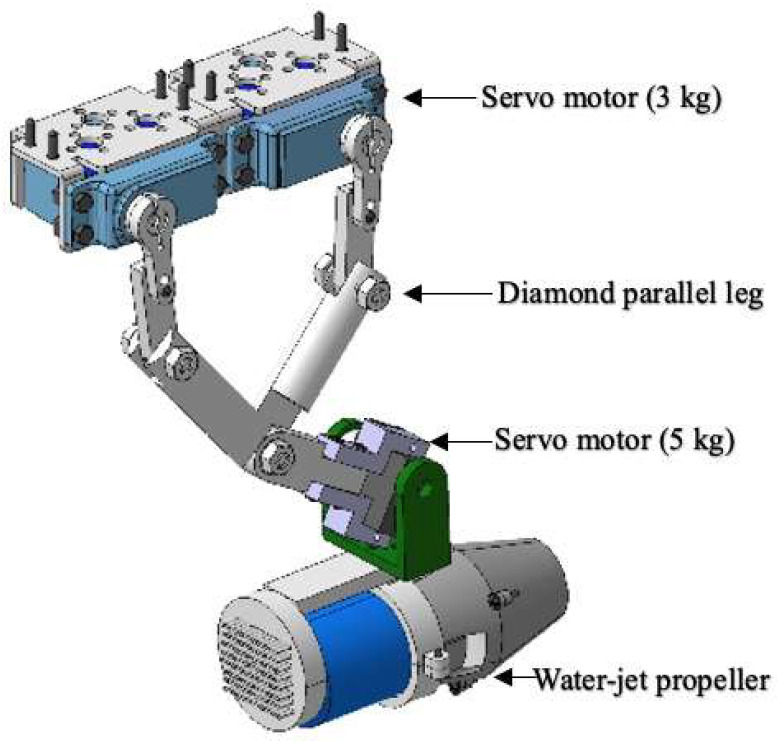
Structure of the diamond parallel leg.

**Figure 4 micromachines-13-02130-f004:**
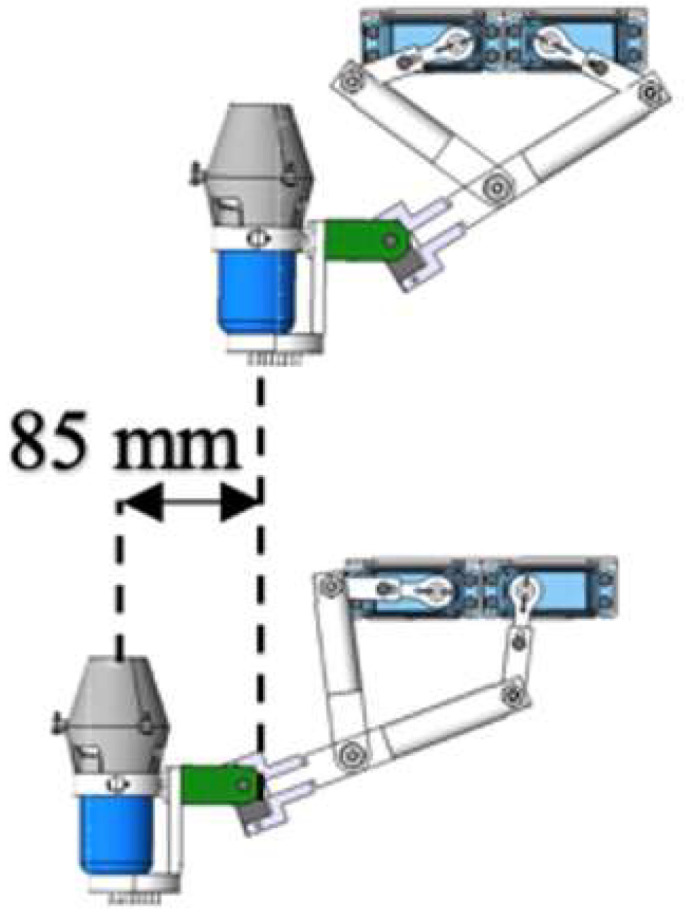
Distance of a step movement.

**Figure 5 micromachines-13-02130-f005:**
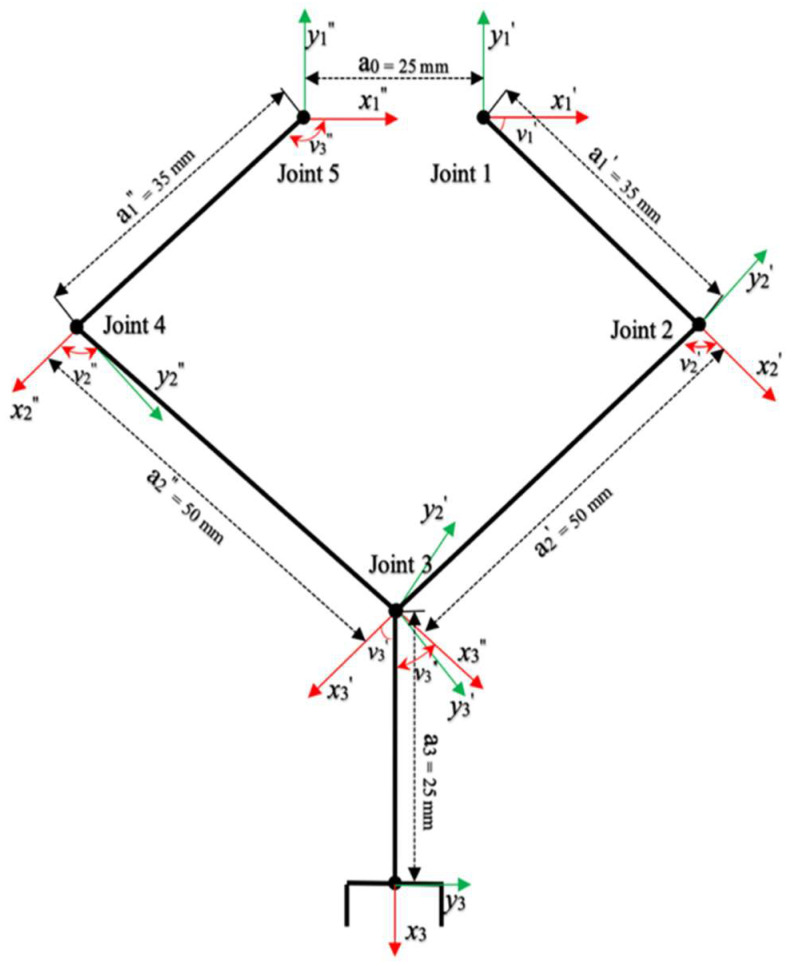
Modelling calibration of the MDH algorithm.

**Figure 6 micromachines-13-02130-f006:**
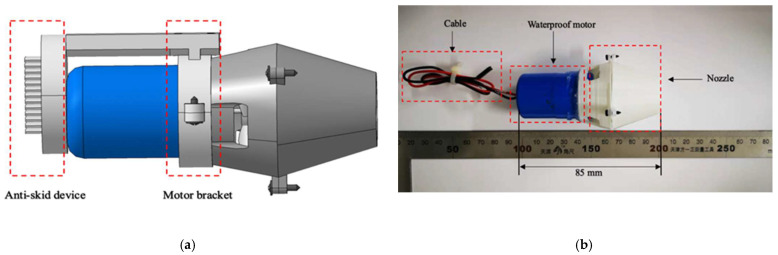
Overall structure of the water jet. (**a**) 3D mechanical structure; (**b**) Overall dimensions.

**Figure 7 micromachines-13-02130-f007:**
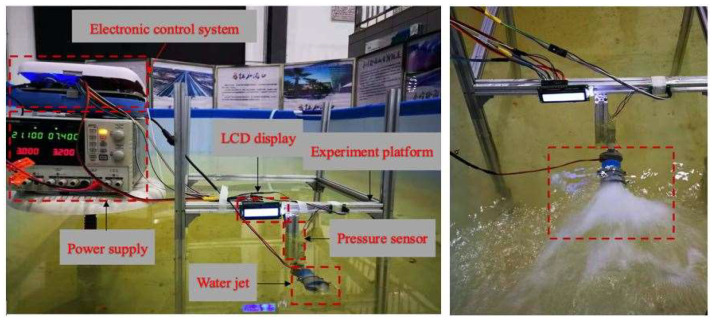
Experimental setup of the water-jet thruster.

**Figure 8 micromachines-13-02130-f008:**
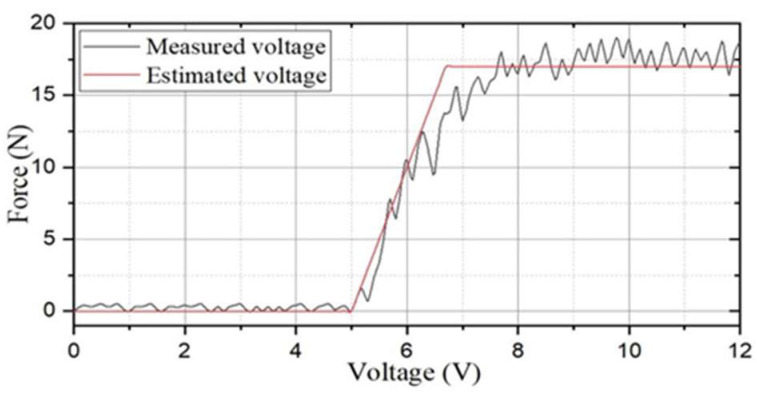
The propulsive force of the different power.

**Figure 9 micromachines-13-02130-f009:**
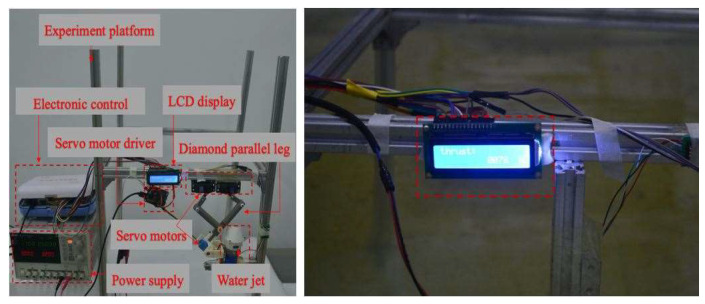
Experimental setup of a diamond parallel leg.

**Figure 10 micromachines-13-02130-f010:**
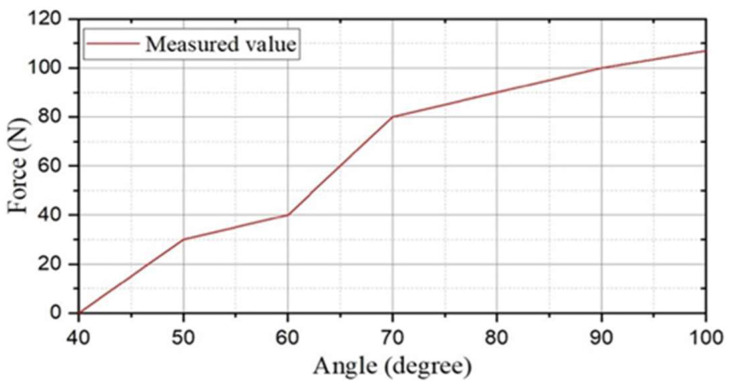
Relationship of the angle and thrust.

**Figure 11 micromachines-13-02130-f011:**
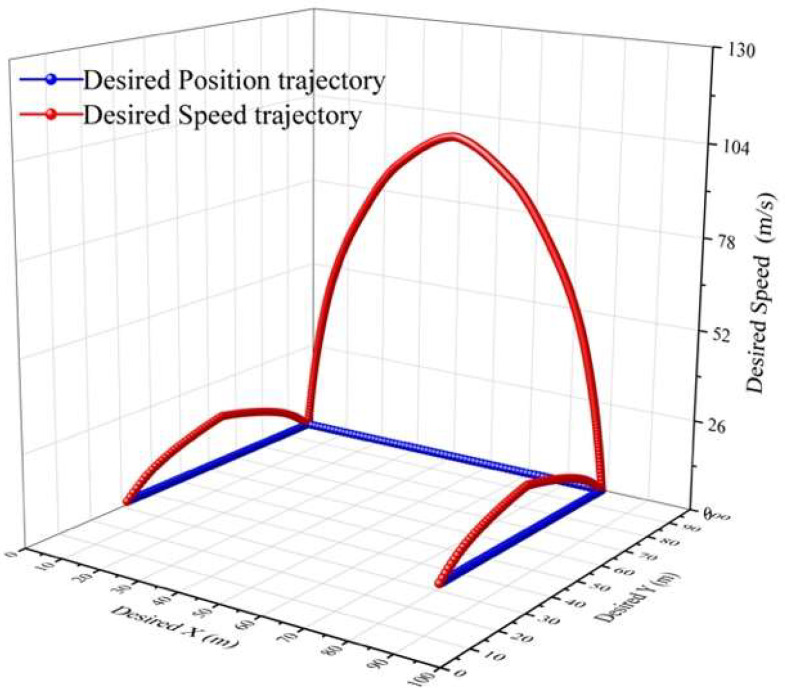
Speed and position trajectory.

**Figure 12 micromachines-13-02130-f012:**
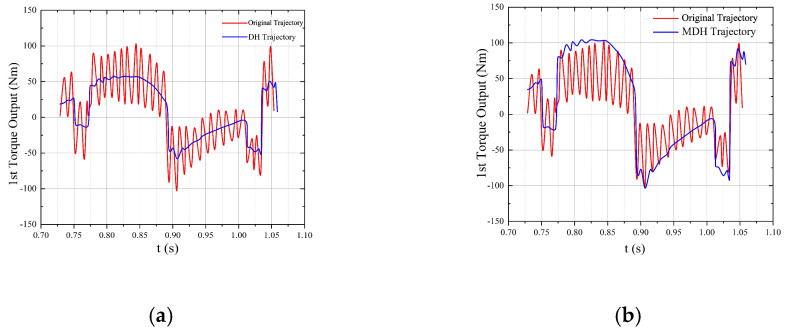
Control behaviours of the DH and MDH control algorithms. (**a**,**b**) torque of the left link; (**c**,**d**) torque of the right link.

**Figure 13 micromachines-13-02130-f013:**
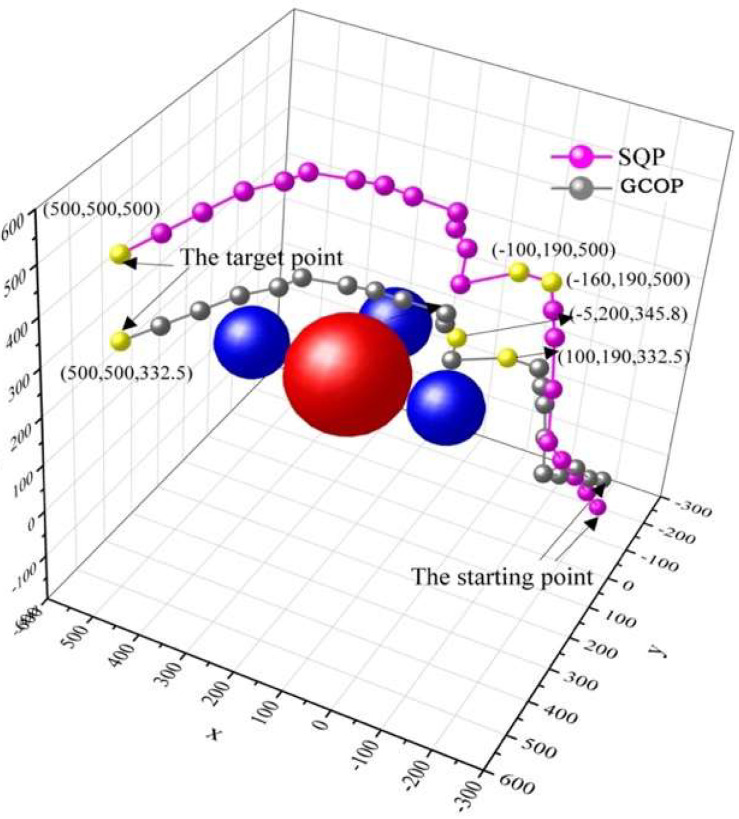
Robot trajectory from *X*-axis path perspective (the red ball represents an obstacle with a diameter = 500 mm and the blue ball represents an obstacle with a diameter = 200 mm).

**Figure 14 micromachines-13-02130-f014:**
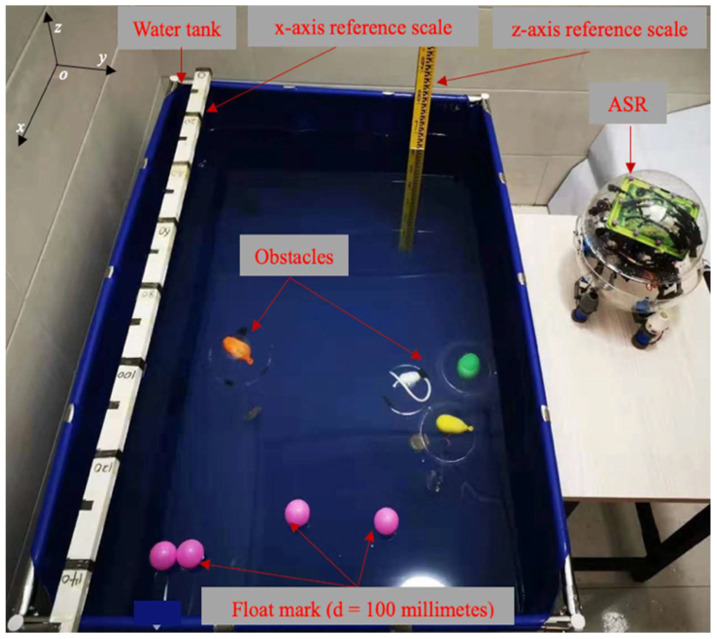
Experimental setup of underwater path-planning.

**Figure 15 micromachines-13-02130-f015:**
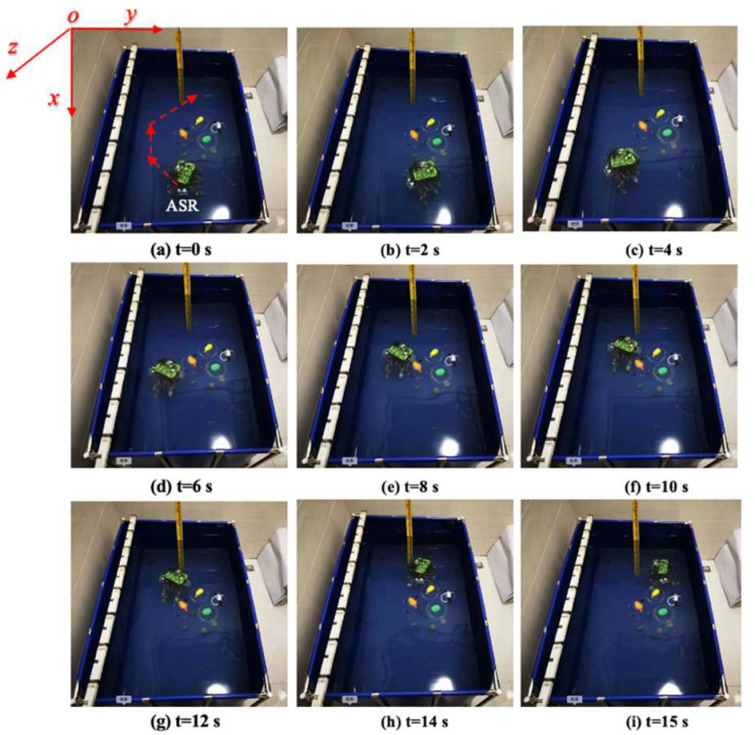
Experimental bypass obstacles.

**Figure 16 micromachines-13-02130-f016:**
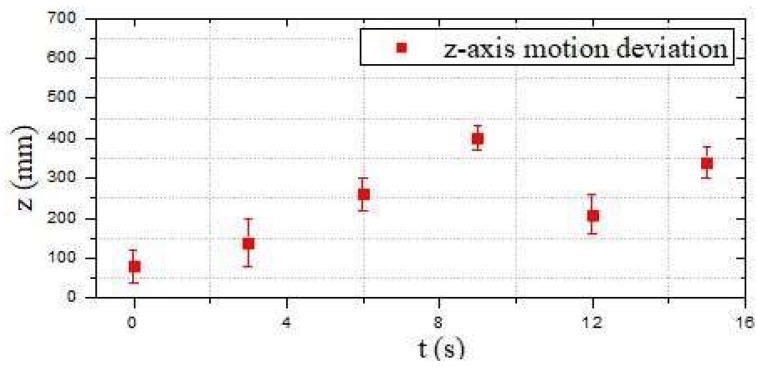
Motion deviation in the *y*-axis direction (the first experiment).

**Figure 17 micromachines-13-02130-f017:**
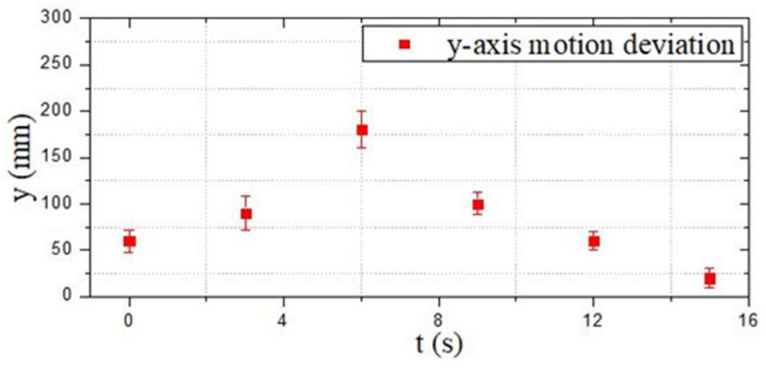
Motion deviation in the *z*-axis direction (the first experiment).

**Figure 18 micromachines-13-02130-f018:**
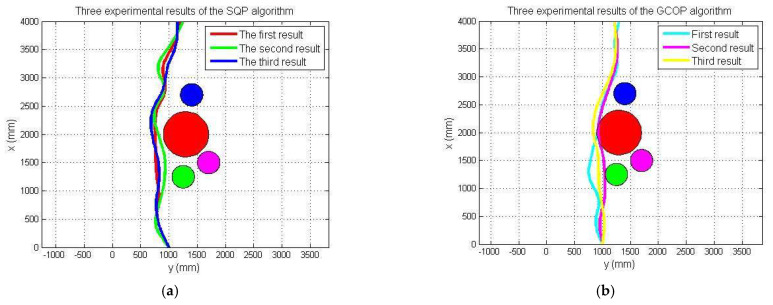
The first experiment of the SQP and GCOP algorithms. (**a**) SOP algorithm; (**b**) GCOP algorithm (the red ball represents an obstacle with a diameter = 500 mm and other balls represents an obstacle with a diameter = 200 mm).

**Figure 19 micromachines-13-02130-f019:**
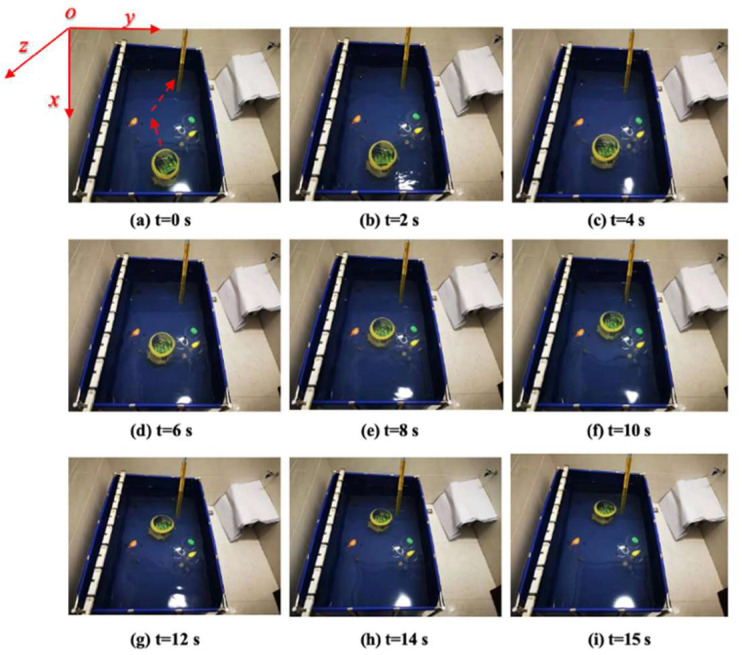
Experiment for passthrough obstacles.

**Figure 20 micromachines-13-02130-f020:**
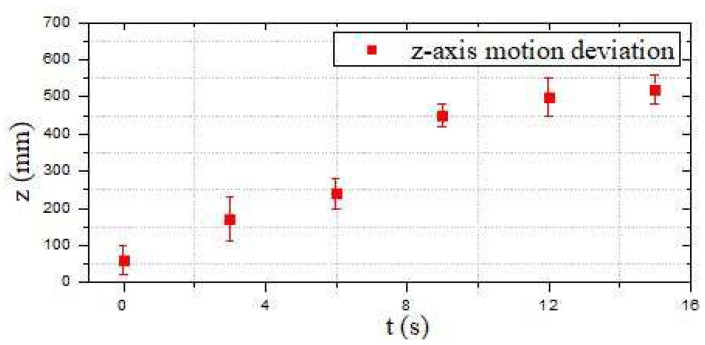
Motion deviation in the *y*-axis direction (the second experiment).

**Figure 21 micromachines-13-02130-f021:**
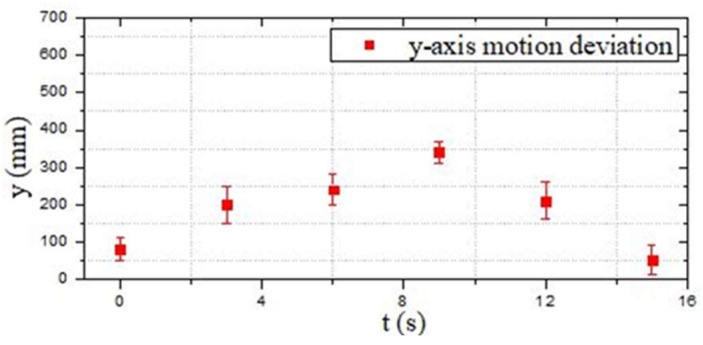
Motion deviation in the *z*-axis direction (the second experiment).

**Figure 22 micromachines-13-02130-f022:**
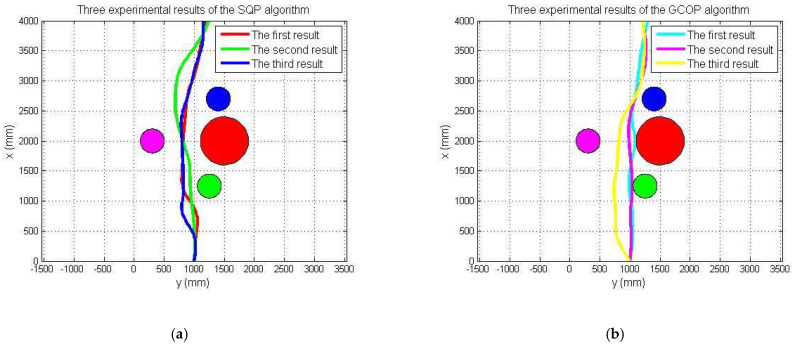
The second experiment of the SQP and GCOP algorithms. (**a**) SOP algorithm; (**b**) GCOP algorithm (the red ball represents an obstacle with a diameter = 500 mm and other balls represents an obstacle with a diameter = 200 mm).

## Data Availability

The authors are unable or have chosen not to specify which data has been used.

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
