# Peer review of "Dynamic Analysis and Path Planning of a Turtle-Inspired Amphibious Spherical Robot"

_micromachines, 2022, doi:10.3390/mi13122130_

Round 1

Reviewer 1 Report

This paper presents the kinematic analysis and path planning of a turtle-inspired amphibious spherical robot. In general, the topic of this paper is interesting but the main contribution/novelty is insufficient. The theoretical contribution is especially inadequate and should be greatly improved. The novelty of the "Path Planning" method should be clearly highlighted. Some comments are as follows.

1. It is claimed "high-precision and high-speed motion control" has been realized. Please clarify this point by using the experimental data, and comparison experiment is necessary. In addition, please indicate the specific criterion/benchmark for the definition of "high-precision" as well as "high-speed".

2. It is claimed "a maximum torque of 102 N" has been realized. But, should the unit of torque be N?

3. The organization of this paper should be improved, and the "kinematic analysis" has not been well covered in the content. Since there is no creative work in kinematic analysis, I think the title of this paper should be reconsidered.

4. The literature research should also be improved. The research gap in this field should be clearly disclosed.

5. The English in this paper still needs to be improved even though it has been proofread. Just as examples, "play an important role in motion control to realize accurate motion control" ? "A dynamic path planning algorithm based on a general constrained optimization problem (GCOP) model and a sequential quadratic programming (SQP) method with sensor input are proposed for the path planning of a turtle-inspired amphibious spherical robot (ASR) in an unknown underwater space to control movement, and thus, achieve collision avoidance." This sentence is too long and is not easy to follow the meaning.

Reviewer 2 Report

In this paper, the authors construct control algorithms that implement the motion of a mobile robot. A Turtle-Inspired Amphibious Spherical Robot, the model of which was proposed in previous works by this group of authors, is used as a full-scale robot model. The paper describes both improvements to the robot itself to make it more maneuverable and stable, as well as modifications to the control algorithms and path planning algorithm. The constructed control algorithms are used to avoid obstacles by a mobile robot.

In general, I believe that this work can be published in Micromachines journal after careful proofreading of the text by the authors and correction of some comments, such as (it just examples):

(1) The authors does not explain what the symbols $c_i$ and $s_i$ mean in equation (3) onwards and a number of other symbols. Authors should carefully check the text for such designations.

(2) Obviously, parametrization (12)--(14) does not lead to identity (11).

(3) It is not clear why the Euler angles $\alpha$, $\beta$, $\gamma$ are mentioned in the sentence following equations (11)--(14), because neither later nor earlier they were in the text.

(4) Check the caption of the figure 11. It seems that last sentence is incomplete or something wrong.

Round 2

Reviewer 1 Report

Most of my concerns have been addressed.